# Tumor Expression of Cyclin-Dependent Kinase 5 (Cdk5) Is a Prognostic Biomarker and Predicts Outcome of Oxaliplatin-Treated Metastatic Colorectal Cancer Patients

**DOI:** 10.3390/cancers11101540

**Published:** 2019-10-11

**Authors:** Vicenç Ruiz de Porras, Sara Bystrup, Sara Cabrero-de las Heras, Eva Musulén, Luis Palomero, Maria Henar Alonso, Rocio Nieto, Diego Arango, Víctor Moreno, Cristina Queralt, José Luis Manzano, Laura Layos, Cristina Bugés, Eva Martinez-Balibrea

**Affiliations:** 1Program of predictive and personalized cancer medicine (PMPPC) Germans Trias i Pujol Research Institute (IGTP), Ctra. Can Ruti—Camí de les escoles s/n, 08916 Badalona, Spain; vruiz@igtp.cat (V.R.d.P.); sbystrup@igtp.cat (S.B.); scabrero@igtp.cat (S.C.-d.l.H.); cqueralt@iconcologia.net (C.Q.); jmanzano@iconcologia.net (J.L.M.); llayos@iconcologia.net (L.L.); cbuges@iconcologia.net (C.B.); 2Program Against Cancer Therapeutic Resistance (ProCURE), Catalan Institute of Oncology, Ctra. Can Ruti- Camí de les escoles s/n, 08916 Badalona, Spain; 3Department of Pathology, Hospital Universitari Germans Trias i Pujol, Ctra. Can Ruti—Camí de les escoles s/n, 08916 Badalona, Spain; emusulen@hotmail.com; 4Department of Pathology, Hospital Universitari General de Catalunya, Grupo Quirónsalud, Pedro i Pons 1, 08195 Sant Cugat del Valles, Spain; 5Program Against Cancer Therapeutic Resistance (ProCURE), Catalan Institute of Oncology, 08908 L’Hospitalet del Llobregat, Barcelona, Spain; lpalomero@iconcologia.net; 6ONCOBELL Program, Bellvitge Institute for Biomedical Research, 08908 L’Hospitalet del Llobregat, Barcelona, Spain; mhalonso@iconcologia.net (M.H.A.); v.moreno@iconcologia.net (V.M.); 7Oncology Data Analytics Program, Institut Català d’Oncologia (ICO), 08908 Barcelona, Spain; 8Consortium for Biomedical Research in Epidemiology and Public Health (CIBERESP), 28029 Madrid, Spain; 9Department of Clinical Sciences, Faculty of Medicine and Health Sciences, University of Barcelona, 08907 Barcelona, Spain; 10Group of Biomedical Research in Digestive Tract Tumors, CIBBIM-Nanomedicine, Vall d’Hebron University Hospital, Research Institute (VHIR), Universitat Autònoma de Barcelona, Passeig Vall d’Hebron, 119-129, 08035 Barcelona, Spain; rocio.nieto@vhir.org (R.N.); diego.arango@vhir.org (D.A.); 11Medical Oncology Service, Catalan Institute of Oncology (ICO), 08908 Barcelona, Spain; 12B-ARGO group, Germans Trias I Pujol Research Institute (IGTP), Ctra. Can Ruti- Camí de les escoles s/n, 08916 Badalona, Spain

**Keywords:** colorectal cancer, cyclin-dependent kinase 5 (Cdk5), prognostic and predictive biomarker, oxaliplatin

## Abstract

In recent years, an increasing number of studies have shown that elevated expression of cyclin dependent kinase (Cdk5) contributes to the oncogenic initiation and progression of many types of cancers. In this study, we investigated the expression pattern of Cdk5 in colorectal cancer (CRC) cell lines and in a large number of tumor samples in order to evaluate its relevance in this pathogenesis and possible use as a prognostic marker. We found that Cdk5 is highly expressed and activated in CRC cell lines and that silencing of the kinase decreases their migration ability. In tumor tissues, Cdk5 is overexpressed compared to normal tissues due to a copy number gain. In patients with localized disease, we found that high Cdk5 levels correlate with poor prognosis, while in the metastatic setting, this was only the case for patients receiving an oxaliplatin-based treatment. When exploring the Cdk5 levels in the consensus molecular subtypes (CMS), we found the lowest levels in subtype 1, where high Cdk5 again was associated with a poorer prognosis. In conclusion, we confirm that Cdk5 is involved in CRC and disease progression and that it could serve as a prognostic and predictive biomarker in this disease.

## 1. Introduction

Cyclin dependent kinase (Cdk5) belongs to the cyclin-dependent kinase family of serine/threonine kinases whose activity depends on a regulatory subunit named cyclin. However, Cdk5 is considered an atypical Cdk because it binds to non-cyclin proteins p35 (*CDK5R1*) and p39 (*CDK5R2*), phosphorylation in the T-loop is not required for its activation and most of its substrates are not involved in cell cycle control. The function of this kinase was thought to be restricted to the central nervous system, where it is indispensable for normal brain development during embryogenesis and controls neuronal migration or axonal guidance, among others. Deregulation of Cdk5 is widely described in neurodegenerative diseases such as Alzheimer’s, Parkinson’s, and Huntington’s [1]. In recent years, several studies have demonstrated that Cdk5 also plays relevant physiological and pathological roles outside of the nervous system, including transcriptional regulation (through regulation of transcription factors such as p53 or signal transducer and activator of transcription 3 (STAT3)), cell migration, adhesion, angiogenesis, and apoptosis [2]. Given that some of these functions are deregulated in tumors and that a great similarity exists between the cellular and molecular mechanisms orchestrating neuronal migration during the development of the nervous system and cancer cell migration during metastasis, it is not surprising that Cdk5 plays a role in cancer. 

Cdk5 phosphorylates the STAT3 transcription factor on its Ser-727 residue, leading to increased proliferation and promotion of tumor formation in medullar thyroid carcinoma and in prostate cancer cells through the activation of androgen receptor [3,4] or resistance to DNA-damaging agents through the increased transcription of the DNA-repair gene essential meiotic structure-specific endonuclease 1 (*EME1)* in colorectal cancer (CRC) cell lines [5]. Cdk5 is involved in the regulation of DNA-damage repair through the phosphorylation of ataxia telangiectasia mutated (ATM) [6]. Elevated expression has been described in human pancreatic [7], lung [8], prostate [9], and breast [10] tumors and has been associated with a worse prognosis. The role of Cdk5 in the proliferation of tumor cells is not clear, but many studies have demonstrated its implication in cancer migration, invasion, and anchorage-independent growth. For instance, in pancreatic cancer, two studies showed that Cdk5 inhibition affected tumor malignant progression in vitro and in vivo. The rat sarcoma (RAS) pathway was reported to have an important effect on the Cdk5-mediated tumor progression on the one hand because mutant Kirsten rat sarcoma (KRAS) increased its kinase activity [7] and on the other hand, by participating in its downstream signaling through phosphorylation of Ras-related (Ral) A and B proteins [11]. In prostate and melanoma cancer models, the inhibition of Cdk5 resulted in decreased cell motility and metastatic potential [9,12] and in breast cancer, Cdk5 controlled cancer cell migration and tumor formation by regulating the phosphorylation of focal adhesion kinase (FAK) at Ser-732, as a downstream step of transforming growth factor Beta (TGF-B1) signaling, and was shown to be essential for epithelial to mesenchymal transition (EMT) and cell motility [10]. In CRC, Cdk5 functions as a tumor promoter via modulating the extracellular receptor kinase 5 (ERK5)—activator protein 1 (AP-1) axis [13] and it has been associated with oxaliplatin resistance acquisition [14].

To date, little is known about the possible role of Cdk5 as a cancer biomarker. In the present study, we studied the role of Cdk5 in several CRC cell lines, reproducing results from Zhuang et al. [13] and most importantly, we investigated the expression patterns of Cdk5 at different molecular levels in a significant number of human tumor samples from patients at different disease stages with the aim of evaluating its role as a prognostic and predictive marker in this malignancy.

## 2. Results

### 2.1. Cdk5 and p35 Are Broadly Expressed in CRC Cell Lines

We studied protein levels of Cdk5 and p35 by western blotting in a panel of 10 human CRC cell lines and detected both at high levels in all cell lines tested (Figure 1A). In contrast, we did not detect the p35 calpain-cleaved product p25, which is recognized by the same antibody. It has been reported that Cdk5 activity is regulated through phosphorylation of Y15 [1], and this phosphorylation is commonly used as a measure of activity [5,15]. However, based on our experience, this is not a good approach because Y15 is a highly conserved residue in the Cdk family [16] and the epitope against which these antibodies are designed is highly homologous for Cdk1, Cdk2, and Cdk5, which could lead to misleading results. As Cdk5 kinase activity is strictly dependent on the binding with p35/p25, this interaction also serves as a measure of activity. To study this binding, Cdk5 was immunoprecipitated and p25 was found to co-immunoprecipitate with Cdk5 in HT29, LoVo, HCT116, and DiFi cell lines (Figure 1B). It was also found that p25 is more stable than p35 and leads to prolonged kinase activity of Cdk5 [17], suggesting a Cdk5 hyperactivation in these cell lines. 

### 2.2. Cdk5 siRNA-Mediated Gene Silencing Impaired CRC Cells Migration and Invasion

We investigated how the transient silencing of Cdk5 affected three important characteristics of tumor cells: proliferation, migration and invasion. Forty-eight hours after siRNA gene silencing, Cdk5 knockdown (siCdk5) efficiency was between 90% and 95% at the protein level in HT29, LoVo, DiFi, and HCT116 cell lines (Appendix A). Under these conditions, we did not observe any effect in cell proliferation (Figure 2A and Appendix A). 

However, we found significantly reduced migration and invasion in HT29 and LoVo siCdk5-transfected cell lines as compared to siRNA non-target control (siNTC) cells. In contrast, silencing of Cdk5 did not decrease the migration and invasion ability of DiFi cells (Figure 2B,C). As HT29 and LoVo cell lines harbor mutations in genes from the mitogen activated protein kinase (MAPK) signaling pathway [18] and the DiFi cell line is considered to be wild type [19], we wondered whether an altered MAPK downstream signaling could be behind the observed differences. We took advantage of another KRAS wildtype (WT) cell line that was engineered to express a mutant form of *KRAS* (G12D). Cdk5 silencing caused a modest but statistically significant decrease in migration only in the SW48 *KRAS* G12D cell line and not in the *KRAS* WT. No differences in invasion ability after Cdk5 silencing were observed in either cell lines (Figure 2D,E).

Another important characteristic of cancer cells is their ability to undergo an unlimited number of divisions. To study the implications of Cdk5 in their long-term clonogenic capability of CRC cells, we performed colony formation assays in the HT29 and LoVo cell lines (Appendix A). We observed a decrease in the number of colonies for siCdk5-transfected cells compared to siNTC cells in both lines, however, these were only statistically significant for the LoVo cells (*p*-values 0.198 and 0.046, respectively). 

### 2.3. Cdk5 and p35 Are Overexpressed in CRC Tumor Samples

We next evaluated Cdk5 and p35 expression in human samples. We included four different cohorts: A, B, D, and F (Table 1). In cohort A, Cdk5 and p35 proteins were clearly overexpressed as detected by western blot in tumor tissues as compared to the corresponding normal adjacent tissues (Figure 3A–C). Cdk5 was detected by immunohistochemistry (IHC) in primary colorectal tumors from cohort D and scored as either negative or positive (weak and strong staining) (Figure 3D). A total of 75% of the tumors were positive for Cdk5 staining. We also observed significantly higher *Cdk5* mRNA levels in tumor versus normal adjacent colon samples in two different publicly available data sets (Colonomics, cohort B, Figure 3E The cancer genome atlas (TCGA), cohort F, Figure 3F). Using TCGA data (cohort F), we analyzed the *Cdk5* gene copy number variation (CNV) (region chr7:151054008–151057848). We observed a statistically significant increase (*p* = 7.58 × 10^−7^) in the Cdk5 copy number in primary tumors as compared to the corresponding adjacent normal colon tissues (Figure 3G). The CNV of the *Cdk5* gene was strongly correlated (r = 0.55, *p* = 7.16 × 10^−7^) to Cdk5 expression levels in tumor tissue (Figure 3H) and also in a panel of 63 CRC cell lines (r = 0.54, *p* = 4.6 × 10^−6^) (Broad Institute Cancer Cell Line Encyclopedia) (Figure 3I).

### 2.4. High Cdk5 Levels Correlate with Poorer Prognosis in CRC Patients 

Taking into account this data, we wondered whether Cdk5 expression could be a good prognostic and/or predictive biomarker in patients. To study this possibility, we used several cohorts in which the expression of Cdk5 was determined through different techniques (Table 1). We categorized data into high and low Cdk5 expression groups as follows: In cohorts using continuous data (cohorts B, C and F), patients were assigned to low or high groups according to Cdk5 values below or above the median, respectively; in those using categorical data (cohorts D and E), patients were grouped as low when IHC staining was negative or weak and as high when it was strong (Figure 3D). The prognostic value of Cdk5 expression was studied in two highly homogenous cohorts of localized microsatellite stable (MSS) CRC cancer patients (cohorts B and C, Table 1). Patients from cohorts B and C were diagnosed with stage II or stage III cancer, respectively, underwent surgery and did not receive any adjuvant treatment. In both the cohorts, high Cdk5 levels were associated with shorter disease free survival (DFS) (*p* = 0.049 95%, confidence interval (CI) 0.98–5.88 for cohort B and *p* = 0.048, 95% CI 0.97–7.43 for cohort C) (Figure 4A,B), and also with overall survival (OS) in cohort B (p = 0.022, 95% CI 0.96–6.48), while no differences could be observed in OS in cohort C (Appendix A). To study the predictive value of Cdk5 expression in the metastatic setting, we used two different, but again, quite homogenous cohorts of patients. Cohort D consisted of stage IV CRC patients treated with oxaliplatin and 5-fluorouracil as the first-line treatment, while patients from cohort E were treated with different schedules combining irinotecan and 5-fluorouracil (Table 1). As shown in Figure 4C,D, high Cdk5 levels were associated with shorter time to progression (TTP) only in the patients that received an oxaliplatin-based treatment (median 8.095 vs. 18.23 months, *p* = 0.043, 95% CI 1.01–4.71) (cohort D, Figure 4C), as no differences could be observed in the patients treated with irinotecan (median 8.95 vs. 8.42 months, *p* = 0.45, 95% CI 0.59–1.27) (cohort E, Figure 4D). These analyses were adjusted by age and sex; however, we performed an additional analysis including metastatic location as another co-variable, which resulted in an increased significance (*p* = 0.018, hazard ratio (HR) = 2.72, 95% CI 1.19–6.23). Similarly, when analyzing the response rates (Chi-square), we only found differences according to Cdk5 levels in the oxaliplatin-treated patients (cohort D) as 87% (13 out of 15 patients) of the low Cdk5 group had complete or partial response, while only 53% (18 out of 34) of the high Cdk5 group responded (*p* = 0.029). In view of these interesting results in the metastatic setting, we wanted to analyze whether Cdk5 was also predictive for adjuvant oxaliplatin-based therapy response. As we do not have our own cohort with this patient group, we used the TCGA data (cohort F). When splitting this subgroup (*n* = 49) based on high or low Cdk5, we could not observe any difference in the progression free interval (PFI) (*p* = 0.64, HR CI) nor OS (*p* = 0.11, HR CI). It is worth mentioning that when analyzing the entire TCGA cohort (*n* = 473, stages I to IV, treated and non-treated), no differences could be observed in PFI and OS (Appendix A), highlighting the importance of choosing appropriate and well-defined cohorts of patients to conduct these studies.

As we observed a difference in migration and invasion when we silenced Cdk5 in our in vitro studies, we analyzed the link between Cdk5 expression and several gene sets related to EMT in the TCGA cohort. EMT is a well described process whereby cells undergo multiple biochemical changes that enable them to assume a mesenchymal phenotype, characterized by enhanced migratory capacity, invasiveness, and elevated resistance to apoptosis. We found that only for the TGF-β signaling within EMT did high Cdk5 levels correlate with higher pathway activation (*p* = 0.026, Appendix A). 

We observed a difference in the effect of Cdk5 silencing in cell lines depending on their *KRAS* mutational status when studying the migration and invasion of the cells. Furthermore, it has been reported that Cdk5 is involved in the signaling cascade downstream of the epithelial growth factor receptor (EGFR) receptor [7]. Thus, we wanted to know whether there was a difference in the prognostic value of Cdk5 according to *KRAS* mutations. Patients from cohort B were split according to *KRAS* mutational status (WT or mutated) and Cdk5 expression was again categorized as high or low. In the *KRAS* WT group, patients with high or low Cdk5 levels had similar DFS (Figure 5A); however, in the group with *KRAS*, mutated high Cdk5 levels predicted a statistically very significant poorer DFS compared to the Cdk5 low group (*p* = 0.004, 95% CI 1.56–36.46) (Figure 5B). We did not see any differences in OS in this cohort when grouping according to the *KRAS* mutational status (Appendix A). 

### 2.5. Cdk5 Is Associated with Consensus Molecular Subtype 1

CRC has recently been classified into four consensus molecular subtypes (CMS) [20], namely, CMS1 (microsatellite instability immune), CMS2 (canonical), CMS3 (metabolic), and CMS4 (mesenchymal) subtypes. The subtype with the worst survival probability is the CMS4 [20]. Taking advantage of the fact that this information was available in cohort F, we wanted to explore whether there was an association between Cdk5 levels and a specific CMS. First, we checked if CMS-associated OS followed the same trend as reported previously and indeed, this was the case (Appendix A). CMS1 displayed the lowest levels of Cdk5 (Figure 6A,B). For each molecular subtype, we then compared the OS between patients with high or low Cdk5 levels and did not observe any differences (Appendix A). When we analyzed the progression-free interval (PFI), an endpoint similar to DFS and TTP used in our previous analysis which has been shown to be the most reliable endpoint when analyzing data from the TCGA-colorectal adenocarcinoma (COAD) database [21], we found that high Cdk5 in the CMS1 correlated with poorer outcomes (*p* = 0.036, 95% CI 1.07–8.10) (Figure 6C), while no differences were found in the three remaining molecular subtypes (Appendix A). It is worth mentioning that the majority of the CMS1 group were non-metastatic patients (55 M0 and 4 M1). 

## 3. Discussion

Cdk5 has emerged as a possible drug target in several tumors, including CRC. Here, we confirm previous results implicating its role in CRC pathogenesis [13], but importantly, we report its possible utility as a prognostic biomarker in this malignancy. It is worth mentioning that our study was performed in a large number of CRC patients’ samples using different techniques to measure Cdk5 levels, and importantly, in selected groups according to tumor stage and clinical approaches. Given the relevance of the recent molecular classification of CRC into four consensus subtypes, we also explored the possible association of Cdk5 overexpression with specific CMS. To our knowledge, this is the most complete study of these characteristics so far.

Our results indicate that Cdk5 and its activator p35 are broadly expressed in human CRC cell lines; however, Cdk5 was found to be mainly bound to p25, the truncated form of p35. The binding of p25 to Cdk5 is indicative of constitutive activation of Cdk5 [17]. In human colorectal adenocarcinomas, Cdk5 and p35 were also highly expressed, especially when compared to normal adjacent tissue or to normal colonic mucosa from healthy donors. Our results are in agreement with those reported by Zhuang et al. [13] and are possibly explained by the increased copy number of the *Cdk5* gene found in primary tumors as compared to normal adjacent tissues, a fact that was also pointed out by Robb et al. [22].

Reduced Cdk5 expression was associated with decreased migration and invasion in vitro but not with cell proliferation. This has been observed previously in cell lines from different tumor origins [7,9,11,12,13]. Our cell lines had mutations in *KRAS* (LoVo) or in rapidly accelerated fibrosarcoma B (*BRAF)* (HT29) genes. Indeed, signaling through the MAPKs is one of the main deregulated pathways in CRC and other authors had reported an interaction between Cdk5 and different effectors of this pathway. Unfortunately, in the only previous study conducted for Cdk5 in the context of CRC, all the cell lines tested had a deregulated MAPK pathway [13]. Therefore, we wanted to include cell lines with proficient MAPK pathways and compare the effects of silencing the *Cdk5* gene. We observed that in this case, Cdk5 downregulation had no effect on migration or invasion of cells, which is in agreement with previously published data [7]. It was previously shown that, in a context of *RAS*/*BRAF* mutant CRC, Cdk5 could phosphorylate ERK5 and that the effects of abrogation of Cdk5 expression were due to the inhibition of ERK signaling [13]. However, we think that this is unlikely as CRC cell lines with *KRAS* or *BRAF* mutations seem to preferentially activate the ERK1/2 pathway and when treated with ERK5 inhibitors, cell proliferation is unaffected [23,24]. 

We could successfully translate our results into a clinical setting as high levels of Cdk5 were associated with a higher risk of relapse in early-stage MSS CRC patients that did not receive any treatment after primary tumor resection, especially in patients with *KRAS*-mutant tumors. These results point to the prognostic value of Cdk5, which seems to be related with a more aggressive phenotype as we and others have reported in cell lines and in vivo models. Thus, those tumors expressing high levels of Cdk5 could have an enhanced capacity to invade and migrate from primary sites to metastatic sites. This is in agreement with what has been reported for other kinds of cancer, for example, breast cancer, where the Cdk5-FAK pathway downstream of TGF-β signaling is necessary for EMT [10]. Indeed, when we analyzed the connection between several gene sets related to EMT and Cdk5 expression in the TCGA cohort, we found that only for TGF-β signaling within EMT did high Cdk5 levels correlate with greater pathway activation. 

We want to remark the homogeneity of these two cohorts of localized CRC patients and the fact that the patients did not receive any treatment, as this underscores the importance of Cdk5 as a biomarker in a priori good prognostic tumors. Whether treatment should be considered for these patients has to be explored in further studies.

We also investigated the predictive value of Cdk5 through analysis of the association between tumor protein expression and the outcome of metastatic CRC patients receiving two standard chemotherapy regimens: Oxaliplatin plus 5-fluorouracil and irinotecan plus 5-fluorouracil. Several studies have shown that Cdk5 plays a role in DNA damage repair and resistance, including the one induced by oxaliplatin and irinotecan [5,14]. However, we only observed a clear association of Cdk5 protein levels with response or TTP in patients treated with the former: the higher the Cdk5 levels, the worse the response or TTP. Whether patients with high Cdk5 tumor levels should be treated with irinotecan-based schedules rather than oxaliplatin-based schedules requires further investigation. Nevertheless, it should be noted that our results were based on the protein assessment in primary tumors and not in metastases. 

For the moment, these results suggest that Cdk5 could be a good target to develop drugs against; on the one hand, they could be used in an adjuvant setting to prevent disease relapse and on the other hand, in a metastatic setting in combination with oxaliplatin. Another possibility would be a combination with poly (ADP-ribose) polymerase (PARP) inhibitors, as it was reported that *Cdk5* gene silencing conferred high sensitivity to this kind of drugs [25]. Unfortunately, although huge efforts have been made, the development of specific inhibitors is still an outstanding issue.

Finally, we wanted to explore the possible association between Cdk5 and the recently described CMSs [20] using the TCGA data. We found that in this cohort, CMS1 tumors had the lowest levels of Cdk5 and a high Cdk5 expression was only associated with worse progression-free interval in this subtype. CMS1 is the so-called immune subtype and it is enriched in microsatellite instable (MSI), hypermutated, and highly lymphocyte-infiltrated tumors, which usually are responsive to immune checkpoint blockers. Dorand et al. reported a CD4+ T-cell-dependent rejection of Cdk5-deficient medulloblastoma cells in mice and an inverse correlation between Cdk5 and CD3+ T-cell infiltration in human medulloblastoma samples [26]. Experimentally, they showed that IFN-γ-induced programmed death ligand 1 (PD-L1) upregulation on medulloblastoma cells requires Cdk5. Therefore, we can speculate that in CMS1 tumors, which display high lymphocyte infiltration, high levels of Cdk5 would favor the expression of PD-L1 avoiding antitumor immunity. Nevertheless, it is important to stress that this cohort is very heterogeneous, containing both localized or metastatic tumors and a variety of treatments. Therefore, this hypothesis should be confirmed in further investigations.

## 4. Material and Methods

### 4.1. Human CRC Cell Lines 

Human tumor-derived colorectal adenocarcinoma cell lines HT29, LoVo, DLD1, LS513, HCT116, DLD1, Caco2, LS174T, SW1417, SW480, HCT15, and DiFi from the American Type Culture Collection, along with SW48 parental and *KRAS* G12D/+ mutated cell lines (Horizon Discovery Ltd, Cambridge United Kingdom), were used in the present study. Cell lines were grown as a monolayer in RPMI 1640 (DLD1, LS513, HCT116, SW48, SW480, Caco2, and HCT15), DMEM (HT29, LS174T and SW1417) and Ham’s F-12 (LoVo) media (Thermo Fisher Scientific, Waltham, MA, USA). DMEM and RPMI media were supplemented with 10% of heat-inactivated fetal calf serum (FCS) (Reactiva, 08004 Barcelona, Spain), 1% penicillin/streptomycin (Thermo Fisher Scientific, Waltham, MA, USA), 2 mmol/L L-glutamine (Sigma-Aldrich, St. Louis, MO, USA), and 10 mmol/L HEPES (Sigma-Aldrich, St. Louis, MO, USA). The Ham’s medium was supplemented with 20% of heat-inactivated FCS and 1% penicillin/streptomycin (Thermo Fisher Scientific, Waltham, MA, USA). All cell lines were cultured at 37 °C in a humidified atmosphere of 5% CO_2_. Cells were periodically tested for *Mycoplasma* contamination and authenticated by short tandem repeat profiling with in-house methods.

The molecular status of cell lines used for experiments: HT29: MSS, *KRAS* WT, and *BRAF* mut. V600E; LoVo: MSI, *KRAS* mut. G13D, and *BRAF* WT; HCT116: MSI, *KRAS* mut. G13D, and *BRAF* WT; DiFi: MSS, *KRAS* WT, and *BRAF* WT; SW48: MSI, *KRAS* WT, and *BRAF* WT.

### 4.2. Gene Silencing

Cdk5 was transiently silenced using a mix of three different siRNAs (s2825, s2826, and s2827; (Thermo Fisher Scientific, Waltham, MA, USA) and lipofectamine RNAiMAX in OptiMem medium according to the manufacturer’s instructions (Thermo Fisher Scientific, Waltham, MA, USA). A negative transcription control (siNTC) (Cat No. AM4611; Thermo Fisher Scientific, Waltham, MA, USA) was used in all the experiments. Silencing was validated by western blotting and experiments with a minimum of 70% silencing levels were considered.

### 4.3. Cell Proliferation

Cell proliferation was analyzed 24, 48, 72, and 96 hours after siRNA Cdk5 silencing. Cells were permeabilized with 0.3% Triton X-100 and 30 μM propidium iodide (Sigma-Aldrich, St. Louis, MO, USA) was added and the fluorescence signal corresponding to the cell quantity was measured at 645 nm on the Varioskan reader (Thermo Fisher Scientific, Waltham, MA, USA). 

### 4.4. Colony Formation Assay

HT29 and LoVo siNTC and siCdk5 cells were plated in six-well plates at a density of 300 cells/well. After 12 days in culture, the cells were washed with phosphate-buffered saline (PBS), fixed with a methanol/acetic acid (3:1) solution for 10 min and stained with a solution of crystal violet (0.5%) for 10 min. Colonies were counted manually.

### 4.5. Migration and Invasion

Migration and invasion assays were performed using transwell plates (HTS Transwell® Sigma-Aldrich, St. Louis, MO, USA). For the invasion assay, the upper chambers were coated with 21 μL of Geltrex™ (Thermo Fisher Scientific, Waltham, MA, USA). For both the assays, the lower chambers were filled with a medium containing 10% FCS and cells were seeded in 2% FCS medium in the upper chamber. Migrated/invasive cells that had reached the lower side of the membrane were fixed with 4% paraformaldehyde (PFA) and stained with 0.5% crystal violet. The ImageJ software was used for quantification.

### 4.6. Immunoprecipitation

Cells were homogenized in lysis buffer and the supernatant was incubated with a Cdk5 primary antibody (Santa Cruz Biotechnology, Santa Cruz, CA, USA; (J3): sc-6247) overnight at 4 °C. The mix was incubated 2 hours with protein G-coupled beads (Millipore, St. Louis, MO, USA) and the beads were washed and boiled for 5 minutes in a loading buffer to detach the protein complex. The protein levels were examined by western blotting. 

### 4.7. Western Blotting

Cells were homogenized in a radioimmunoprecipitation assay (RIPA) plus buffer. Frozen tissue samples were disintegrated manually and were homogenized with RIPA plus buffer using the gentleMACS Dissociator system (Milteny Biotech, Bergisch Gladbach, Germany). Protein concentration was determined using the DC™ Protein Assay (Bio-Rad Laboratories, Inc., Richmond, CA, USA) and 50 μg of the protein was loaded and subjected to electrophoresis in 10% Sodium dodecyl sulfate-polyacrylamide gel electrophoresis (SDS-PAGE) gels (Thermo Fisher Scientific, Waltham, MA, USA) and transferred onto Polyvinylidene fluoride or polyvinylidene difluoride (PVDF) membranes (Bio-Rad Laboratories, Inc., Richmond, CA, USA). After blocking (LICOR Biosciences, Lincoln, NE, USA), membranes were incubated overnight with specific primary antibodies against Cdk5 (Cell Signaling, Danvers, MA, USA, #2506, 1:1000), p35 (Cell Signaling, Danvers, MA, USA, #2680, 1:300), and α-Tubulin (Sigma-Aldrich, St. Louis, MO, USA, #T6074, 1:20,000). Membranes were incubated with IRDye rabbit and mouse secondary antibodies (1:10,000) (LICOR Biosciences, Lincoln, NE, USA) and scanned and analyzed on the Odyssey imaging system (LICOR Biosciences). Non-cropped western blots from Figure 1B, Figure 3A, and Appendix A can be seen in Appendix A.

### 4.8. Patients’ Samples 

Six different cohorts (named A to F), with a total of 811 samples from different patients, were used in this study. Table 1 shows the characteristics of each cohort. Samples from cohort C were collected at the Duran i Reynals Hospital. Samples from cohorts D and E belong to a private collection (ISCIII registered number C.0001505). All the samples were collected according to standard local protocols and after obtaining informed consent. The Clinical Research Ethical Committee from Hospital Germans Trias i Pujol provided approval for the study (BB14002; date: 7th March 2014). 

### 4.9. Tissue Microarray and Immunohistochemistry (IHC) Staining

A tissue microarray (TMA) was built for cohorts D and E as previously described [27]. Subsequent immunohistochemistry procedures were applied as reported [27,28], and specific Cdk5 antibodies were used (Abcam, Cambridge, MA, USA, ab40773, 1:200 and Cell Signaling, Danvers, MA, USA, 2506, 1:200 for validation). Cells with siNTC or siCdk5 gene silencing were used to evaluate antibody specificity. 

### 4.10. qPCR

*CDK5* gene expression was studied by qPCR in cohort C as previously described [14,28]. Briefly, retrotranscription was performed with moloney murine leukemia virus (MMLV) reverse transcriptase (Thermo Fisher Scientific, Waltham, MA, USA) and the Cdk5 assay no. Hs00762869_s1 was used (Thermo Fisher Scientific, Waltham, MA, USA). Relative gene expression quantification was calculated according to the comparative Ct method as described elsewhere using β-Actin (Thermo Fisher Scientific, Waltham, MA, USA) as the endogenous control.

### 4.11. In Silico Data

Two publicly available cohorts were used: Colonomics (NCBI BioProject PRJNA188510) and TCGA. The colonomics cohort (cohort B) consisted of 98 paired adjacent-normal and tumor tissues from stage II microsatellite stable patients and 50 colon mucosae samples from healthy donors (246 samples in total). Gene expression data, assessed by Affymetrix Human Genome U219 expression array, had previously been analyzed [29]. 

For studies using the TCGA cohort (cohort F), we included all samples annotated as TCGA-colon adenocarcinoma (COAD) (*n* = 448) and 25 samples annotated as stage IV TCGA-rectum adenocarcinoma (READ), 473 samples in total. The classification of TCGA samples into four CMS was in agreement with Guinney et al. [20]. Different survival event data were retrieved according to Liu et al. [21]. 

### 4.12. Statistical Analysis

The data coming from in vitro experiments are presented as mean ± standard error of the mean (SEM) of at least three independent experiments and statistical analysis was performed with Graphpad Prism V.4 software (San Diego, CA, USA). Comparisons among different experimental conditions were carried out through the T-student or two-way ANOVA test followed by Bonferroni post-test. Values of *p* ≤ 0.05 were considered significant. 

A survival analysis was performed with the survival and survminer R packages. Kaplan–Meier curves depicting disease-free survival (DFS), time to progression (TTP), progression-free interval (PFI), and overall survival (OS) were accompanied with the log-rank test to verify significance in survival curve differences. The Cox regression hazards models were performed to quantify the effect of gene expression and survival (adjusted for age and stage). Differences in response to treatment were evaluated in cohorts D and E by contingency tables and chi-square or Fisher’s exact tests as appropriate. 

Regarding TCGA data, comparison between primary and normal tumor samples (gene expression and copy number alteration segment mean) were performed using paired samples T-tests. Both variables were compared using the Pearson correlation test. *Cdk5* gene expression comparison between different CMS subtypes was performed using one-way ANOVA. 

The GSVA R-package (gene set variation analysis, PMC3618321) was used to compare the gene expression levels of EMT and TGF-β signaling in EMT pathways in Cdk5 low vs. Cdk5 high tumors. A Wilcox’s test was performed to evaluate the statistical significance.

## 5. Conclusions 

We combined in vitro studies with the analysis of a high number of tumor samples from different CRC stages and confirmed that Cdk5 is involved in CRC progression and that it could serve as a prognostic and predictive biomarker in this disease. Our findings also suggest that Cdk5 could help decide which patients should receive adjuvant therapy and whether oxaliplatin or irinotecan should be used in the treatment of metastatic disease. 

## Figures and Tables

**Figure 1 cancers-11-01540-f001:**
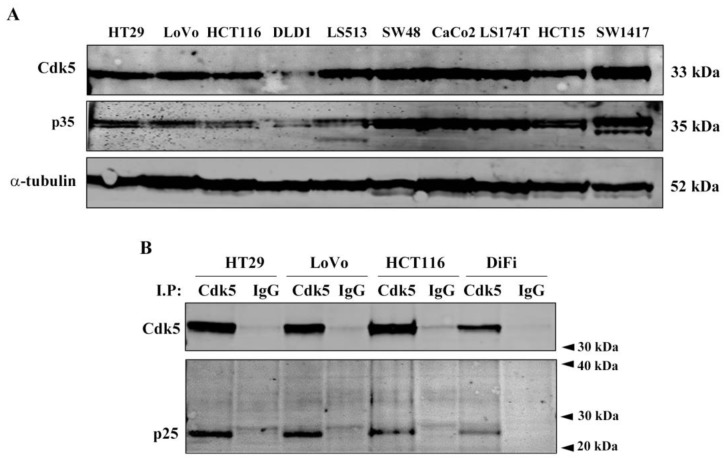
Cyclin dependent kinase 5 (Cdk5) and p35 (CDK5R1) expression in colorectal cancer (CRC) cell lines. (**A**) Western blot analysis of Cdk5 and p35 basal expression in a panel of 10 CRC cell lines. Alpha-tubulin was used as endogenous control. (**B**) Representative western blot images showing the co-immunoprecipitation of Cdk5 and p35/p25 in the indicated cell lines. Co-immunoprecipitation with an immunoglobin G (IgG) antibody was used as a negative control. The results were obtained from at least three independent experiments.

**Figure 2 cancers-11-01540-f002:**
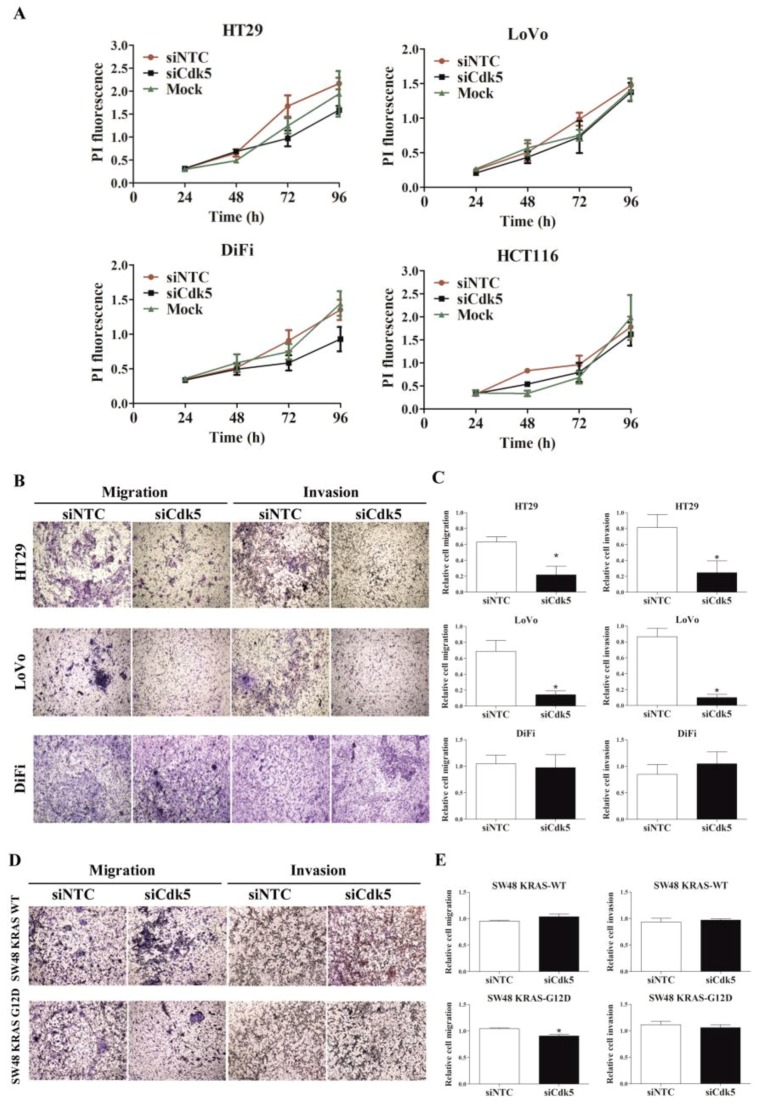
Effect of Cyclin dependent kinase 5 (Cdk5) siRNA-mediated gene silencing on colorectal cancer (CRC) cells proliferation, migration and invasion. (**A**) Graphic representation of HT29, LoVo, DiFi, and HCT116 time-dependent cell proliferation after *Cdk5* gene silencing measured by propidium iodide (PI). (**B**,**D**) Representative Boyden chamber migration and invasion assays images (4× magnification) and (**C**,**E**) bar graphs showing (mean ± SD) relative cell migration and invasion after *Cdk5* silencing in the indicated cell lines. * *p*-value < 0.05 relative to control cells siRNA non-target control (siNTC). The results were obtained from at least three independent experiments.

**Figure 3 cancers-11-01540-f003:**
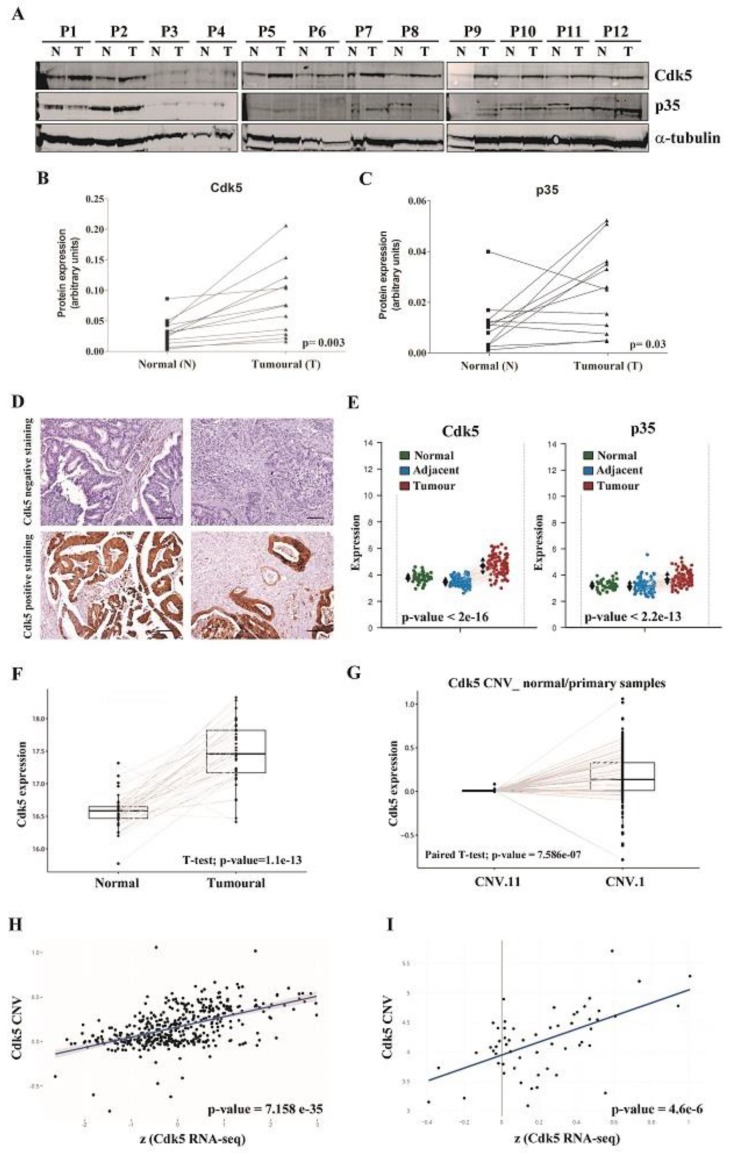
Cyclin dependent kinase (Cdk5) and p35 expression in colorectal cancer (CRC) tumor samples. Western blot (**A**) and graphic representation (**B**,**C**) of Cdk5 and p35 protein expression, respectively, in tumoral (T) and normal adjacent (N) tissues of 12 stage IV CRC patients (cohort A). Alpha-tubulin was used as endogenous control. The *p*-value was according to paired *t*-test. (**D**) Representative immunohistochemistry images of Cdk5 staining in CRC tumor tissues. The upper panel shows negative staining and lower panel positive staining. Scale bar: 100 μm. (**E**) Graphic representation of *Cdk5* and *p35* mRNA expression in 98-paired adjacent normal and tumoral tissues from stage II microsatellite stable (MSS) CRC patients (cohort B). (**F**) Graphic representation of *Cdk5* mRNA expression in normal and tumoral tissues of 38 I-IV CRC patients. Data were obtained from The Cancer Genome Atlas (TCGA) database (cohort F). (**G**) Graph representing the *Cdk5* copy number (CNV) change between normal and tumoral tissues in 67 stage I–IV CRC patients (cohort F). (**H**) Correlation between *Cdk5* CNV and *Cdk5* gene expression in 429 stage I–IV CRC patients. p-value according to Pearson correlation test (cohort F). (**I**) Correlation between *Cdk5* CNV and *Cdk5* gene expression in 63 CRC cell lines. The *p*-value was according to Pearson correlation test. Data from the Broad Institute Cancer Cell Line Encyclopedia.

**Figure 4 cancers-11-01540-f004:**
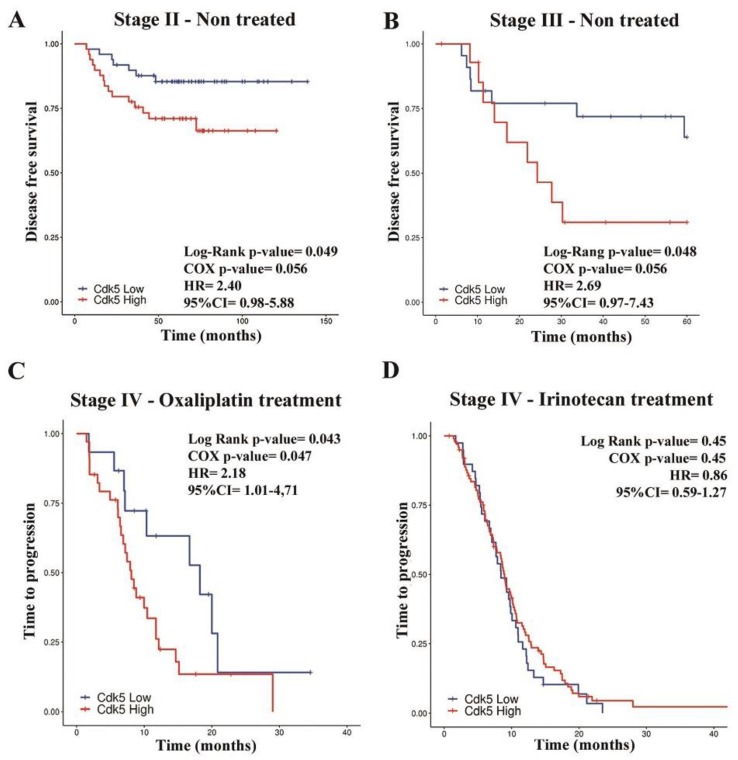
Kaplan–Meyer analysis of disease free survival (DFS) and time to progression (TTP) depending on Cyclin dependent kinase (Cdk5) levels. (**A**) DFS in 98 stage II colorectal cancer (CRC) patients split by the median of Cdk5 expression (cohort B). (**B**) DFS in 37 non-treated stage III CRC patients split by the median of Cdk5 expression (cohort C). (**C**) TTP in 52 stage IV oxaliplatin-treated patients, grouped depending on negative (*n* = 18) or positive Cdk5 (*n* = 34) immunohistochemistry (IHC) staining (cohort D). (**D**) TTP in 139 stage IV irinotecan-treated patients grouped depending on negative (*n* = 73) and positive (*n* = 66) Cdk5 IHC staining (cohort E).

**Figure 5 cancers-11-01540-f005:**
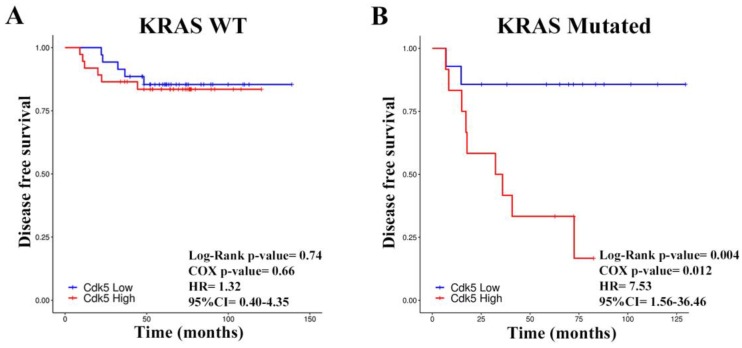
Kaplan–Meyer analysis of disease free survival (DFS) depending on Cyclin dependent kinase (Cdk5) expression and Kirsten rat sarcoma oncogene (*KRAS)* mutational status. (**A**) DFS for 72 stage II colorectal cancer (CRC) patients with wildtype (WT) *KRAS* and split by the median of Cdk5 expression. (**B**) DFS for 26 stage II CRC patients with mutated *KRAS* and split by the median of Cdk5 expression (cohort B).

**Figure 6 cancers-11-01540-f006:**
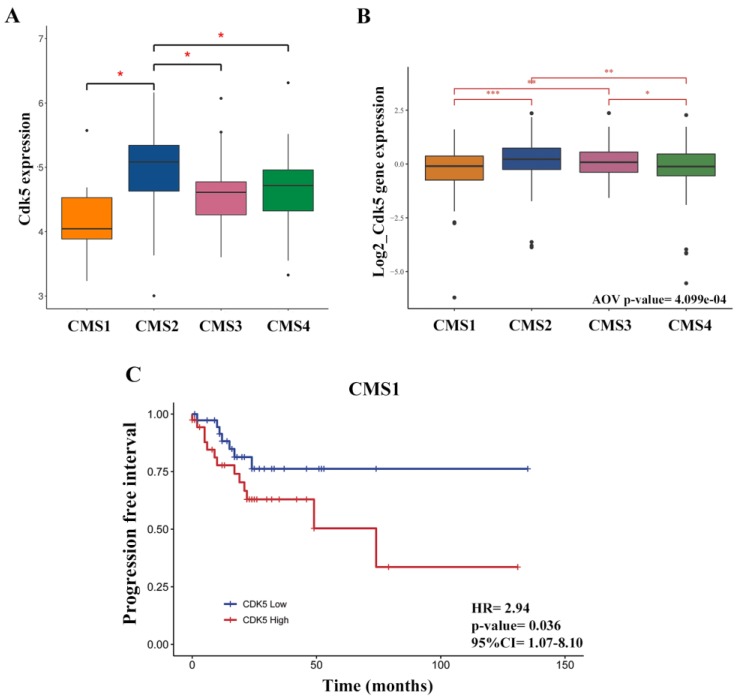
Cyclin dependent kinase 5 (Cdk5) expression in the four consensus molecular subtypes (CMS). (**A**) Dunn and Kruskal–Wallis multiple comparison of Cdk5 expression in CMS in 98 stage II tumors. Note that only six cases were classified as CMS1 as this cohort was restricted to microsatellite stabile (MSS) cases (**B**) ANOVA analysis of Cdk5 expression levels in the CMS in the cancer genome atlas - colorectal adenocarsinoma - rectal adenocarcinoma dataset (TCGA-COAD-READ) including 410 stage I–IV patients. * *p*-value < 0.05, ** *p*-value < 0.01, *** *p*-value < 0.001. (**C**) PFI in the TCGA-COAD CMS1 subgroup including 78 stage I–IV patients; patients were split according to the median of Cdk5 expression (cohort F).

**Table 1 cancers-11-01540-t001:** Overview of cohort used in the study.

Cohort	Type of Samples	Cdk5 Measurement	N	Sex	Age	Metastasis	Stage	Treatment	Origin of Samples	Available at	Molecular Data Available	Clinical Data Available
**A**	Frozen tissue Tumor/adjacent	WB	12	Male 7 (78%) Female 2 (22%)	72	Liver 8 (89%) Lung 1 (11%)	IV	N/A	Tumor Biobank	N/A	-	-
**B**	In silico data	Micro array	98	Male 72 (57%) Female 28 (43%)	72 (43-87)	-	II, MSS	Radical surgery 96 (98%)	Colonomics project	Colonomics.org	RAS, CMS, etc	DFS, OS
**C**	Frozen tissue	qPCR	37	Male 21 (57%) Female 16 (43%)	78 (37-91)	-	III	Radical surgery 36 (97%)	Duran and Reynals Hospital	N/A	-	DFS (mean 36.3 months) OS (mean 42 months)
**D**	FFPE – TMA	IHC	52	Male 29 (56%) Female 23 (44%)	62 (37-76)	Liver 37 (71%) Lung 5 (10%) Others 10 (19%)	IV	5-FU/OXA (77%) CAPE/OXA (23%)	Private collection	ISCIII	-	DFS (mean 9.6 months)
**E**	FFPE – TMA	IHC	139	Male 78 (69%) Female 35 (31%)	62 (29-75)	Liver 97 (70%) Lung 46 (33%) Others 20 (14%)	IV	5-FU/LV/IRI (47%) 5-FU/IRI (53%)	Private collection	ISCIII	-	DFS (mean 9.4 months) OS (mean 20.3 months)
**F**	In silico data	RNA seq	473	Male 259 (54%) Female 225 (46%)	66 (31-90)	-	I-IV	Various	TCGA project	https://cancergenome.nih.gov	CNV, RAS, CMS, etc	PFI, OS

Abbreviations: 5-FU/LV/IRI: 5-Fluorouracil/leucovorin/irinotecan; 5-FU/IRI: 5-Fluorouracil/irinotecan; 5-FU/OXA: 5-Fluorouracil/oxaliplatin; CAPE/OXA: Capecitabine/oxaliplatin; CMS: Consensus molecular subtype; CNV: Copy number variation; DFS: Disease-free survival; FFPE-TMA: Formalin-fixed paraffin-embedded tissue microarray; IHC: Immunohistochemistry; ISCIII: Instituto de Salud Carlos III; MSS: Microsatellite stable; N/A: Not available; OS: Overall survival; PFI: Progression-free interval; qPCR: Quantitative polymerase chain reaction; TCGA: The cancer genome atlas; RAS: Rat sarcoma oncogene; WB: Western blot.

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
