# Peer review of "Tumor Expression of Cyclin-Dependent Kinase 5 (Cdk5) Is a Prognostic Biomarker and Predicts Outcome of Oxaliplatin-Treated Metastatic Colorectal Cancer Patients"

_cancers, 2019, doi:10.3390/cancers11101540_

Round 1

Reviewer 1 Report

This paper explores the functional ramifications of Cdk5 expression in tumour progression and aims to highlight the significance of Cdk5 expression as a prognostic and predictive marker in CRC. While the novelty in this paper lies in the predictive nature of Cdk5 in oxaliplatin response, the first two sections of the results section are essentially replicating findings reported in Zhuang 2016 Cell Death and Disease. This part is therefore not novel and not as well presented as in the previous paper. 

-line 53: Cdk5 not cdk5 (capitalise C for consistency)

-line 67: "prognosis" not "prognostic"

-line 76: need to be consistent in describing phosphorylation (see line 61)

-line 95: "SW48" should read "HCT116"

-Figure 1B: remove the ladder on the right and replace it with markers with size indicators. 

-Figure 2C: change y-axis of HT29 for migration to 1.0 for consistency

Figure 3D: Add scale bar to image

-The TCGA cohort is only used in a pooled analysis of stage 1-IV. Do your findings by stage and prediction for oxaliplatin occur in the TCGA cohort?

-Are the differences in survival seen in Stage II and III CRC between low and high Cdk5 expression different in MSI versus MSS?

-What do you propose is an explanation for the differences observed in invasion in the KRAS mutant cell lines and a lack of reproducibility in the SW48 mutant knockout?

Reviewer 2 Report

The current manuscript describes a novel role of Cdk5 in CRC malignancy. The authors provide a reliable set of evidence obtained from several cell culture models as well as from different patient cohorts. Their key findings suggest a role of Cdk5 stimulating migration/invasiveness/EMT in the context of cells bearing mutations in the MAPK pathway. In vivo evidence points to an association between higher Cdk5 levels and poorer prognosis. This effect was dependent on the type of treatment applied and the consensus molecular subtype. With help of different types of cohorts, the authors were able to find several associations that usually remained undiscovered. Although further research is still required, these findings suggest Cdk5 as a potential biomarker. The manuscript is well written, the experimental design is appropriate and the conclusions are properly supported by the experimental evidence. I recommend only some (mostly minor) changes to improve readability and clarity of the article.

Major points

Figure 3I is missing in the version for peer review and was therefore not evaluated.

Minor points

Please, provide clone or product numbers for all antibodies used in western blot studies. Fig 1A. To clearly show that the band of p25 is absent in the p35 representative image, the position of the molecular weight marker needs to be indicated. Otherwise, readers may not really know where the p25 band should have appeared. Is there any rational for displaying cell proliferation data for HCT116 cells as supplementary information, while the corresponding curves for HT29, LoVo and DiFi cells are shown in the main manuscript (Fig. 2a)? If possible, I suggest displaying the 4 cell lines together in Fig 2. Section 4.5. To improve readability, could you better distinguish between the experimental procedure applied for the migration and the invasion assay? Figure S3: Please, check labeling of the panels in the figure legend. Panel D is mentioned twice. Results, section 2.5. A reference to figure 6A is missing in the text. Results, lines 229-230. The statement associating high Cdk5 expression and poorer outcome in CMS1 is only displayed in Fig 6C. A statement describing results from FigS5B should be added.

Reviewer 3 Report

In this paper, Vicenç Ruiz de Porras et al evaluated Cdk5 as prognostic biomarker in colorectal cancer (CRC) but also as predictor of oxaliplatin-based chemotherapy. It is an important clinical topic to identify new prognostic/predictive biomarkers in CRC. Moreover this study combined both pre-clinical and clinical data which is great and allows having a mechanistic explanation. The manuscript is well written. Nevertheless, there are some major issues.

Major limitations

- In the Introduction/Discussion: authors should described what is new as compared to the following paper “CDK5 functions as a tumor promoter in human colorectal cancer via modulating the ERK5-AP-1 axis”.

- The results concerning the function of Cdk5 according to KRAS status is not enough strong (see Fig 2E) to conclude that “silencing of the kinase decreases the migration and invasion in cell lines harboring 34 mutations in the MAPK signaling pathway”. Results must be confirmed by using others cell lines.

- Since Cdk5 is predictive of oxaliplatin-based chemotherapy efficacy in mCRC, it should be interesting to evaluate the predictive value of oxaliplatin-based chemotherapy efficacy in non-metastatic CRC in a recent population of stage III CRC treated with oxaliplatin-based chemotherapy.

- Concerning Cdk5 as predictor of oxaliplatin-based chemotherapy efficacy in mCRC there are some issues: p value is limit (p=0.043), patient’s number is low (n=52) and patient’s number in each subgroup must be provided, the gold standard is PFS but not TTP and no multivariate analysis taking into account others prognostic criteria has been performed (i.e. age, performance status..). It should be great to validate these results in a larger series of mCRC treated with oxaliplatin-based chemotherapy and see if Cdk5 remain predictive biomarker in multivariate analysis.

- Concerning the link between TGF- β signaling and Cdk5 levels, one more time, the correlation is low and seems not different when looking at Fig S4 since confidence intervals intersect.

- PFI is not defined and not a good endpoint, DFS, PFS and OS are the gold standards.

- Concerning the association between high level of Cdk5 and poor prognosis in CMS1 it is difficult to interpret the results since all stage are mixed and numbers of patients under the survival curves are not provided. Analyses must be performed in M0 and M1 CRC separately and patient’s number provided.

Minor Essential Revisions

- In the Introduction functions of p35 should be explained.

- Why the cells lines used changed according the experiments? All results should be together in all different cell lines including HCT116 (not in Supp data).

 - Please provide western blot with better quality.

- There is neither legend nor definition of abbreviations used in different tables.

- All abbreviations should be defined (RIPA, SEM…) and/or explained (TCGA-COAD).

- “These results point out the prognostic value of Cdk5, which seems to be related with a more aggressive phenotype as we and others have reported in cell lines and in vivo models. Thus, those tumors expressing high levels of Cdk5 could have enhanced capacity to invade and migrate from primary sites to metastatic sites. Once in the metastatic site it is possible that having high or low Cdk5  levels does not confer any advantage or disadvantage to the cancer cells, a fact that would explain the lack of association between OS and Cdk5 levels” : these sentence are completely speculative since invasion/migration are both necessary at non-metastatic and metastatic stages.

- More data are necessary before to say “Therefore, patients with high Cdk5 tumor levels should not be treated with oxaliplatin but with irinotecan-based schedules”.

Round 2

Reviewer 1 Report

Thankyou for addressing my concerns and altering the manuscript accordingly to address these issues. 

Author Response

Thank you for the thorough revision of the manuscript.

Kind regards

Reviewer 3 Report

The authors have read my comments carefully and have tried to answer all the questions. This new version of the paper improves the relevance of the results and clarifies different aspects of this work. Nevertheless, some issues remain to be access; especially some answers to my comments are relevant but not included in the revised manuscript:

- Mention that Cdk5 is not predictive of oxaliplatin-based chemotherapy efficacy in non-metastatic CRC (see A3).

- Mention that 22 additional cases that were treated with cetuximab+FOLFOX were analyzed and the significance is maintained (see A4).

- Mention that results in CMS1 is based on 55 M0 CRC and 4 M1 CRC and then the results and conclusions are relevant only for non-metastatic CRC  (see A7).

- Say the molecular status of each cell lines used in the Methods (RAS, BRAF and MSI) (see A9).

Authors should explain what is presented in the Figure S4: min – max, median, and quartiles… (see A5).

I am not sure that it is relevant to use PFI that combined TTP and DFS and mixed metastatic and non-metastatic CRC patients. The editor needs to decide if it is ok for him or not (see A6). In addition editor should decide if more cell lines are necessary or not (see A9) and if new western blot with “better quality” are necessary or not (see A10).

After doing these modifications, I hope the paper is acceptable for publication.

Best regards

Author Response

Please see the attachment for a point by point answer to the last comments. 
